# Oxidative Stress as a Target for Non-Pharmacological Intervention in MAFLD: Could There Be a Role for EVOO?

**DOI:** 10.3390/antiox13060731

**Published:** 2024-06-16

**Authors:** Aurelio Seidita, Alessandra Cusimano, Alessandra Giuliano, Maria Meli, Antonio Carroccio, Maurizio Soresi, Lydia Giannitrapani

**Affiliations:** 1Unit of Internal Medicine, “V. Cervello” Hospital, Ospedali Riuniti “Villa Sofia-Cervello”, Department of Health Promotion Sciences, Maternal and Infant Care, Internal Medicine and Medical Specialties (PROMISE), University of Palermo, 90146 Palermo, Italy; aurelio.seidita@unipa.it (A.S.); alegiuliano94@gmail.com (A.G.); maria.meli@unipa.it (M.M.); antonio.carroccio@unipa.it (A.C.); 2Institute for Biomedical Research and Innovation (IRIB), National Research Council (CNR), 90146 Palermo, Italy; alessandra.cusimano01@unipa.it; 3Unit of Internal Medicine, University Hospital “P. Giaccone”, Department of Health Promotion Sciences, Maternal and Infant Care, Internal Medicine and Medical Specialties (PROMISE), University of Palermo, 90127 Palermo, Italy; maurizio.soresi@unipa.it

**Keywords:** ROS, oxidative stress, MAFLD, MD, EVOO

## Abstract

Oxidative stress plays a central role in most chronic liver diseases and, in particular, in metabolic dysfunction-associated fatty liver disease (MAFLD), the new definition of an old condition known as non-alcoholic fatty liver disease (NAFLD). The mechanisms leading to hepatocellular fat accumulation in genetically predisposed individuals who adopt a sedentary lifestyle and consume an obesogenic diet progress through mitochondrial and endoplasmic reticulum dysfunction, which amplifies reactive oxygen species (ROS) production, lipid peroxidation, malondialdehyde (MDA) formation, and influence the release of chronic inflammation and liver damage biomarkers, such as pro-inflammatory cytokines. This close pathogenetic link has been a key stimulus in the search for therapeutic approaches targeting oxidative stress to treat steatosis, and a number of clinical trials have been conducted to date on subjects with NAFLD using drugs as well as supplements or nutraceutical products. Vitamin E, Vitamin D, and Silybin are the most studied substances, but several non-pharmacological approaches have also been explored, especially lifestyle and diet modifications. Among the dietary approaches, the Mediterranean Diet (MD) seems to be the most reliable for affecting liver steatosis, probably with the added value of the presence of extra virgin olive oil (EVOO), a healthy food with a high content of monounsaturated fatty acids, especially oleic acid, and variable concentrations of phenols (oleocanthal) and phenolic alcohols, such as hydroxytyrosol (HT) and tyrosol (Tyr). In this review, we focus on non-pharmacological interventions in MAFLD treatment that target oxidative stress and, in particular, on the role of EVOO as one of the main antioxidant components of the MD.

## 1. Introduction

Recently, the definition of non-alcoholic fatty liver disease (NAFLD) has been changed to steatotic liver disease (SLD) [1], a new classification that better defines the spectrum of steatotic pathology, also allowing better prognostic evaluations [2]. According to the new nomenclature, facing a new diagnosis of steatosis, we confront different scenarios, including metabolic dysfunction-associated fatty liver disease (MAFLD), a condition in which steatosis is associated with those metabolic dysfunctions typical of metabolic syndrome (MetS), type 2 diabetes mellitus (T2DM), arterial hypertension, visceral obesity (high waist circumference values), hypertriglyceridemia, and low HDL levels, all conditions related to insulin resistance (IR); MAFLD associated with alcohol intake (MetALD) at varying degrees—between 20–50 g/day in women and 30–60 g/day in men; alcohol-related liver disease (ALD), with an intake of >50 g/day in women and >60 g/day in men; or, finally, specific etiology SLD. 

Putting all these aspects of a multifaceted condition together, the general consensus has been to underline the importance of non-modifiable (age, sex, race/ethnicity, family history, genetics) and modifiable (lifestyle/diet/exercise, comorbidities, drugs, alcohol) risk factors in the natural history of SLD, which is quite variable [3].

Moreover, despite the widespread diffusion of SLD, there are still no drugs approved for any specific treatment, in particular, that are capable of blocking the evolution of non-alcoholic steatohepatitis (NASH) toward advanced stages of fibrosis and its resulting complications.

However, in addition to those involving drugs, several clinical trials have been performed to test the efficacy of nutritional supplements, such as vitamins, as a complementary therapy in its management, and, in particular, non-pharmacological approaches targeting oxidative stress have been widely studied.

## 2. Oxidative Stress and Liver Steatosis

Oxidative stress plays a crucial role in the pathophysiology of NAFLD. It occurs because of an imbalance between the production of reactive oxygen species (ROS) and antioxidant activity [4]. Reactive nitrogen species (RNS), which are ROS with nitrogen instead of oxygen, also cause oxidative stress [5].

ROS originate during metabolic processes like cellular respiration, fatty acid oxidation, physical exercise [6], and immune response. ROS are also generated when the organism metabolizes xenobiotic substances, such as alcohol or air pollutants [7,8], though moderate levels of ROS are essential for several cellular processes, such as cell development, apoptosis [9], phagocytosis [10], and stimulation of initial immune system response [11]. ROS include superoxide radical anion (O_2_•^−^), hydroxyl radical (OH•), hydrogen peroxide (H_2_O_2_), nitric oxide (NO•), nitrogen dioxide (NO_2_•), and, when elevated into cells, react with biomolecules causing damage, breaking DNA strands, and attacking cell membranes and fatty acids, causing lipid peroxidation [12]. Some enzymes in cell membranes create ROS, such as NADPH oxidase, which appears to have an essential role in the development of NAFLD as it produces O_2_•^−^ from H_2_O_2_, increasing oxidative stress [13].

Oxidative stress can lead to endoplasmic reticulum stress in the liver and increase the release of inflammatory cytokines and macrophages. These two conditions activate hepatic stellate cells (HSCs) that secrete collagen, leading to fibrosis [14,15].

Although the causes of NAFLD are multiple, oxidative stress is an essential factor in the development of the disease, especially in the transition from NAFLD to NASH [16]. NAFLD is characterized by an initial phase of accumulation of liver lipids caused by IR and a high-fat diet, along with genetic predisposition [17]. At the same time, hyperinsulinemia occurs, increasing lipolysis in peripheral tissues with a consequent increase in free fatty acids that are deposited as triglycerides in the liver. This condition leads to the onset of inflammatory states, metabolic disorders, and excessive activity of the mitochondria with increased ROS [18].

In NAFLD patients, high levels of ROS trigger lipid peroxidation of polyunsaturated fatty acids (PUFAs), causing an increase in highly reactive compounds such as thiobarbituric acid reactive substances (TBARSs), malondialdehyde (MDA), and F2α-isoprostane, which are markers of lipid peroxidation and oxidative stress [4,12]. High ROS levels also regulate the inhibition of nuclear factor erythroid 2-related factor 2 (Nrf2), which promotes the transcription of antioxidant genes [4,19]. Among the antioxidant defense systems, the activity of several enzymes is essential. For example, superoxide dismutases (SODs) catalyze the dismutation of O_2_•^−^ into H_2_O_2_; catalase (CAT) reduces two molecules of H_2_O_2_ into two H_2_O and one O_2_; glutathione peroxidases (GPXs) are a family of peroxidases that catalyze the conversion of H_2_O_2_ into H_2_O and O_2_ using glutathione (GSH) as an electron donor, converting it to the oxidized form glutathione disulfide (GSSG). In humans, there are eight isoforms of GPXs, the most abundant of which is GPX-1, present in most tissues [5]. Glutathione reductase also plays a central role in cellular redox homeostasis. Indeed, this enzyme catalyzes the reduction of GSSG to GSH. GSH is one of the most abundant and important antioxidant thiols in humans, modulating ROS and RNS levels [20].

Different methods allow the estimation of ROS levels and oxidative damage, such as directly measuring ROS content, even though they have a short lifespan because of their instability. Antioxidant status can be measured, in particular, enzymatic antioxidant activity (SOD, catalase, and GPXs), non-enzymatic antioxidant activity (GSH), and total antioxidant activity. Furthermore, it is possible to indirectly measure oxidative stress by evaluating the levels of damage byproducts (TBARS, MDA) [21]. Many studies have demonstrated decreased antioxidant systems and increased ROS in NAFLD patients, a condition that exacerbates liver injury [21,22] (Figure 1).

## 3. The Role of Vitamin E and Vitamin D Supplementation

### 3.1. Vitamin E Supplementation in the Treatment of Liver Steatosis over the Decades: Where Do We Stand?

One of the vitamins most frequently tested so far for NAFLD therapy is vitamin E, which, by limiting membrane injury from ROS generated in the liver, should theoretically reverse ROS actions in the recruitment of activated inflammatory cells, impairment of nucleotide and protein synthesis, injury of organelles and activation of HSCs, and impairment of membrane function, including insulin signaling [23], with consequent biological benefits.

Vitamin E is a fat-soluble vitamin that exists in eight variants: α, β, γ, or δ tocopherol and α, β, γ, or δ tocotrienol. The most biologically active form is α-tocopherol, which is often used as a synonym for vitamin E. Vitamin E is a powerful antioxidant with free radical scavenger activity, avoiding lipid peroxidation, but it also acts by increasing the activity of antioxidant enzymes like SOD and GPx [24,25,26,27]. In addition to antioxidant activity, vitamin E also has an anti-inflammatory effect by the following: inhibiting COX2 and 5-LOX [28], anti-tumor activity [29], protective effects against cardiovascular events [30], and regulating the immune response [31].

Vitamin E’s antioxidant role in NAFLD and NASH has been widely studied. Indeed, Phung et al. demonstrated the beneficial effect of vitamin E (250 mg/kg diet) on oxidative stress in a mouse model fed a high-fat and methionine choline-deficient (MCD) diet to induce steatohepatitis. After 10 weeks of vitamin E supplementation, hepatic GSH increased and hepatic TBARS levels decreased compared to the MCD mice, showing its scavenging and antioxidant activity. Moreover, vitamin E also reduced inflammation and steatosis compared to MCD mice but also reduced HSC activation, hepatic collagen α1 mRNA, and fibrosis, thereby ameliorating liver injury [32].

Nan et al. demonstrated that vitamin E (100 mg/kg chow) and 1-aminobenzotriazole (100 mg/kg), another antioxidant, reduced hepatic MDA concentration and enhanced hepatic SOD activity in mice fed an MCD diet after 10 days and 3 weeks. Inflammation was also reduced by vitamin E and 1-aminobenzotriazole treatment; in fact, COX-2 mRNA and protein and nuclear protein NF-kB in the liver decreased. Antioxidants limited liver fibrosis modulation of mRNA as well as TGF-β1 and MMP2 protein expression, which decreased with treatment [33].

Another study also corroborated the previous research. In particular, rats fed a high-fat and high-cholesterol diet to create a NAFLD model showed decreased SOD activity and increased MDA content, a condition that vitamin E (75 mg/kg day) treatment for 8 weeks reverted [34].

It is known that exercise is also essential for treating liver steatosis, together with healthy food and nutraceutical supplementation. Indeed, Bai et al. demonstrated that rats with NAFLD (induced with a high-fat diet; HFD rats) treated with vitamin E and aerobic exercise (VE + E + HFD group) showed reduced ROS and MDA levels, increased SOD and GPx levels compared to HFD rats, HFD rats treated with exercise, and HFD rats treated with vitamin E only. Of note, even vitamin E supplementation alone had a potent antioxidant effect. The VE + E + HFD group showed a lower body weight than other groups; in fact, the treatment also regulated lipid metabolism through increased adenosine monophosphate-activated protein kinase (AMPK). pAMPK blocks carnitine palmitoyl-transferase 1C (ACC), inhibiting fatty acid synthesis. Moreover, regulatory element-binding protein-1 (SREBP-1), which regulates triglyceride synthesis, decreased, and carnitine palmitoyl-transferase 1C (CPT1C), which promotes β-oxidation, increased [35].

Furthermore, vitamin E protects against hepatic oxidative stress by reducing the liver GSSG/GSH ratio, decreasing liver TBARS, and the expression levels of mRNAs of genes involved in pro-oxidant activity, such as liver NADPH oxidase 2 (Nox2), as demonstrated by Presa et al. in a NAFLD mouse model. Endoplasmic reticulum stress is linked to oxidative stress; in fact, it is also attenuated by vitamin E treatment, reducing the protein levels of CCAAT-enhancer-binding protein homologous protein (CHOP), binding immunoglobulin protein (BiP), and protein disulfide-isomerase (PDI). A lower inflammatory state and less fibrosis after vitamin E supplementation were observed. In fact, Tnfα and Cd68 mRNA, two inflammatory markers, decreased, as did alpha-1 type I collagen (Col1a1) and metalloproteinases 1 (Timp1), two markers of fibrosis [14].

### 3.2. Clinical Trials

In the clinical trials published utilizing this therapeutic approach, vitamin E was administered over various numbers of years, at different doses, and as monotherapy or in combination therapy. Reviewing data from the literature, it seems that they differ mostly according to the population of NAFLD patients studied, with pediatric populations less likely to experience therapeutic benefits than adults. In fact, at least in the first studies, a pilot protocol by Lavine et al. published in 2000 showed that, in 11 obese children, vitamin E (doses from 400 to 1200 IU/d) caused major improvements in liver enzyme levels, independently of weight loss, though the benefits were lost after the withdrawal of vitamin E [36]. A subsequent study on a child population with obesity-related liver dysfunction compared the effect of vitamin E and weight loss on transaminase values and on ultrasonographic bright liver, finding that the variations in transaminase levels and the percentage of patients with normalized transaminase values were comparable in the two groups [37]. These findings were later corroborated by Nobili et al. who, comparing the effects of a nutritional program alone or in combination with alpha-tocopherol and ascorbic acid on ALT levels and IR in biopsy-proven NAFLD children, found that diet and physical exercise led to significant improvement of liver function and glucose metabolism, beyond any antioxidant therapy [38]. Also, Lavine et al., in a randomized, double-blind, placebo-controlled clinical trial of 173 patients (aged 8–17 years) with biopsy-confirmed NAFLD, showed that the daily administration of 800 IU of vitamin E (58 patients), 1000 mg of metformin (57patients), or placebo (58 patients) for 96 weeks, was not superior to placebo in attaining a sustained reduction in ALT level [39]. In the same year, Akcam et al. arrived at similar conclusions studying sixty-seven obese adolescents with liver steatosis (age range, 9–17 years) who were randomized either to receive metformin at a dose of 850 mg daily or vitamin E at a dose of 400 U daily, plus individually tailored diet, exercise, and behavioral therapy for 6 months. At the end of the study, the authors found that metformin treatment was more effective than dietary advice and vitamin E treatment in reducing IR and ameliorating metabolic parameters such as fasting insulin and lipid levels [40].

However, more recently published trials have highlighted the possibility of improvements in the oxidative stress parameters in obese prepubertal children with liver steatosis treated either with lifestyle modifications combined with Vitamin E [41] or with a combination of docosahexaenoic acid–choline–vitamin E [42] or, finally, with a mixture of vitamin E and hydroxytyrosol (HT) [43,44]. 

In the adult population, one of the first pilot studies performed on subjects with NASH showed that only lifestyle modifications (low-fat diet and exercise) were associated with a significant improvement in liver enzymes, whereas the vitamin E supplementation used in this short-term study provided no apparent added benefit [45]. Similar results were achieved in a randomized controlled trial of metformin versus vitamin E or prescriptive diet in NAFLD, which proved that metformin treatment was better than a prescriptive diet or vitamin E in the treatment of NAFLD patients receiving nutritional counseling [46].

However, a prospective, double-blind, randomized, placebo-controlled trial of 45 histologically-proven NASH patients who were randomized to receive either vitamins E and C (1000 IU and 1000 mg, respectively) or placebo daily for 6 months showed at their histological evaluation that the vitamin treatment group experienced a statistically significant improvement in fibrosis score (FS), while no changes were noted in inflammation scores [47].

Moreover, in 2004, Sanyal et al. published a randomized prospective trial performed to compare the efficacy and safety of vitamin E alone (400 IU/day) vs. vitamin E (400 IU/day) and pioglitazone (30 mg/day) in subjects with NASH. The histological evaluation performed after six months from baseline showed that the combination of vitamin E and pioglitazone produced a greater improvement in NASH histology. The authors hypothesized that the improvement in steatosis and cytologic ballooning were related to treatment-associated decreases in fasting FFA and insulin levels. However, treatment with vitamin E only also produced a significant decrease in steatosis [48]. A wider case study by the same authors on NASH patients studied 247 adults who were randomly assigned to receive pioglitazone at a dose of 30 mg daily (80 subjects), vitamin E at a dose of 800 IU daily (84 subjects), or placebo (83 subjects), for 96 weeks. At the end of the observation period, the vitamin E therapy group, compared with the placebo group, was associated with a significantly higher rate of improvement in NASH, but the difference in the rate of improvement with pioglitazone compared to placebo was not significant; serum ALT/AST levels were reduced with vitamin E and pioglitazone, compared to placebo, and both agents were associated with reductions in steatosis and lobular inflammation, but not with improvement in FS [49].

In 2006, Dufour et al. performed a randomized placebo-controlled trial of ursodeoxycholic acid (UDCA) with vitamin E in NASH. They enrolled 48 patients with biopsy-proven NASH who were randomly assigned to have UDCA 12–15 mg/kg per day with vitamin E 400 IU twice per day (UDCA/Vit E), UDCA with placebo (UDCA/P), or placebo/placebo (P/P) and were followed for two years, after which they underwent a second liver biopsy. At the end of the follow up, serum AST/ALT levels were significantly diminished in the UDCA/Vit E group, while only ALT levels did so in the UDCA/P group. Neither AST nor ALT levels improved in the P/P group. For histology, the activity index was significantly better in the UDCA/Vit E group, mostly as a result of steatosis regression, while it was unchanged in the P/P and UDCA/P groups [50]. Moreover, besides the effects of UDCA/VitE on aminotransferase levels and liver histology in patients with NASH, a decrease in hepatocellular apoptosis markers and a restoration of circulating levels of adiponectin were observed by Balmer et al., results which suggested that the UDCA/VitE combination could have metabolic effects in addition to its beneficial cytoprotective properties [51]. 

Similar conclusions regarding the positive effects of Vitamin E in NASH at the histological level were reached by Hoofnagle et al. They assessed how changes in ALT elevations could reflect improvements in liver histology in response to vitamin E therapy, showing that decreases in ALT were more frequent among vitamin E recipients than in the placebo group and that they were also associated with decreases in NAFLD activity score (NAS) but not in FS [52].

More recently, numerous other studies have been conducted with various combinations of other compounds rather than vitamin E alone. In particular, a combination with another antioxidant, alpha lipoic acid (ALA), was tested in a placebo-controlled, open-label, prospective study of 155 patients with NAFLD and NASH who were randomized for treatment with ALA 300 mg, vitamin E 700 IU, ALA 300 mg plus vitamin E 700 IU, or placebo daily for 6 months. The authors found that treatments of ALA and vitamin E alone or in combination improved inflammatory cytokine levels, steatosis scores, homeostasis model assessment scores, and triglyceride levels [53].

In 2015, Aller et al. tested the efficacy of the administration of a combination of silymarin plus vitamin E, together with a lifestyle modification program (hypocaloric diet and exercise), in comparison with only the hypocaloric diet, in a 3-month study of a group of 36 biopsy-proven NAFLD patients. At the end of the study, they observed a significant reduction in gamma-glutamyltransferase (GGt) levels, fatty liver index (FLI), liver accumulation product (LAP), and NAFLD-FS in both groups, but only in the treated group did these values remain significant even in subjects who did not reach 5% weight loss [54]. The oral administration of a symbiotic and vitamin E combination was proposed as an effective treatment in 60 NAFLD patients who were recruited in a randomized, double-blind, controlled clinical trial, which showed significant efficacy in reducing liver enzyme levels and metabolic parameters, thus demonstrating a possible synergistic effect of multiple strains of probiotic and vitamin E supplements in NAFLD patients [55].

The combined administration of vitamin E and pioglitazone was further tested by Bril et al., who performed a randomized, double-blind, placebo-controlled trial, including 105 patients with T2DM and biopsy-proven NASH who were randomly assigned to vitamin E 400 IU b.i.d., vitamin E 400 IU b.i.d. plus pioglitazone 45 mg/day, or placebo groups. Both treated groups showed improvements in the resolution of NASH compared with placebo; steatosis improved with combination therapy and vitamin E alone, though inflammation and ballooning only improved with combination therapy, and no improvement in fibrosis was observed in any group [56].

Another randomized double-blind, placebo-controlled trial was conducted by Anushiravani et al. on 150 consecutive patients with NAFLD, who were assigned to lifestyle plus placebo, metformin 500 mg/day, silymarin 140 mg/day, pioglitazone 15 mg/day, and vitamin E 400 IU/day groups, all for 3 months. Liver enzymes and anthropometric parameters decreased significantly in the treatment groups, without any significance among the treatments [57].

### 3.3. Does Hypovitaminosis D Play a Causal Role in Liver Steatosis Development?

Vitamin D is a fat-soluble vitamin that is partly taken in through food but also synthesized by the human body. In the skin, 7-dehydrocholesterol is transformed by UV radiation into cholecalciferol and then metabolized in the liver and kidneys into its active form, 1,25-dihydroxycholecalciferol. Vitamin D regulates the metabolism of calcium and phosphorus and is, therefore, essential for the skeletal system; it also has antioxidant, anti-inflammatory, and immunomodulatory activity [58].

These properties have been fully recognized, but a nontraditional role of vitamin D has also been reported in cancer and autoimmune disease [59], as well as in chronic liver disease and chronic hepatitis C patients, where vitamin D is involved in regulating the immune system, inflammatory response, and fibrogenesis [60].

Recent studies show vitamin D’s involvement in improving oxidative stress. For example, Reda et al. demonstrated that vitamin D supplementation (1000 IU/kg BW) given 3 days/week for 10 weeks in NAFLD rats decreased hepatic MDA levels, while SOD and GSH increased. Vitamin D treatment also reduced inflammatory status, increasing IL-10 and decreasing NF-kB protein levels. SREBP-1c was downregulated and PPARα upregulated, demonstrating that vitamin D decreased lipogenesis and increased β-oxidation, improving lipid metabolism [61].

In another study, authors used rats fed a choline-deficient (CD) diet to induce NASH (CD group) and then administered vitamin D at different dosages (1, 5, or 10 μg/kg of body weight). After 12 weeks of treatment, vitamin D ameliorated oxidative stress in the NASH rat group with vitamin D supplementation, decreasing hepatic levels of TBARS compared to the CD group but not affecting total antioxidant capacity (TAOC). Treatment with vitamin D also improved fibrosis; in fact, hepatic TIMP-1 and α-smooth muscle actin (α-SMA), correlated with the degree of fibrosis, were downregulated. Furthermore, the higher dose of vitamin D did not seem to bring improvements [62].

The antioxidant effect of vitamin D has also been demonstrated in an in vivo model of MetS. HFD rats were treated with vitamin D, metformin, or both for 8 weeks. In all three groups, serum MDA levels were lower than in untreated HFD rats [63].

Moreover, Ma et al. found that vitamin D supplementation ameliorated oxidative stress in an in vivo model of NAFLD. They administered different doses of vitamin D to NAFLD rats (1, 5, 10 µg/kg). After 12 weeks, vitamin D treatment reduced liver MDA and ROS levels as well as the expression of TBARS, while TAOC increased, especially with the 5 µg/kg dose. The lower doses of vitamin D (1 µg/kg and especially 5 µg/kg) also reduced steatosis and fibrosis, as demonstrated through histopathology analysis, and decreased liver levels of IL-2, IL-6, and TNFα, alleviating inflammation [64].

In recent years, nanoemulsions have been widely used to improve the delivery of drugs and compounds. El-Sherbiny et al. have demonstrated the anti-inflammatory and antioxidant effects of a vitamin D nanoemulsion (obtained by the sonification and change in pH of pea protein extract and canola oil) in HFD rats. Its antioxidant activity is mediated by a hepatic Nrf2 mRNA level that is higher in vitamin D-treated HFD rats compared to HFD rats but much higher in vitamin D nanoemulsion-treated HFD rats. The latter also downregulated serum levels and liver gene expression of TNFα and increased serum IL-10 levels compared to the HFD and vitamin D alone groups. Furthermore, hepatic steatosis and fibrosis decreased with vitamin D nanoemulsion supplementation. Cpt1a mRNA expression level was upregulated, and lipids were downregulated in the liver, demonstrating that vitamin D nanoemulsion also improved lipid metabolism [65].

In contrast, Zhu et al. found that Nrf2 mRNA did not significantly increase in HFD rats and vitamin D-treated HFD rats compared to control. However, they found positively stained Nrf2 by immunohistochemistry in the nucleus, but not in the cytosol, in vitamin D-treated-HFD rats. Vitamin D promotes Nrf2 translocation from the cytosol to the nucleus, allowing the expression of antioxidant genes. Indeed, vitamin D-treated-HFD rats showed high levels of expression of CAT, SOD2, glutamate-cysteine ligase catalytic subunit (GCLC), and NAD(P)H–quinone acceptor oxidoreductase 1 (NOQ1) mRNAs, which are known for their antioxidant activity. Furthermore, in vitamin D-treated-HFD rats, hepatic MDA and F2α-isoprostane decreased. Histopathological analysis showed a reduction in steatosis and fibrosis compared to the HFD group, as well as a decrease in lipid droplets in the liver, supporting the ameliorative effect of vitamin D treatment on lipid metabolism and the development of NASH [66].

Palmitic acid (PA) treatment caused fat accumulation and inflammation in murine AML-12 hepatocytes, increasing ROS levels and lipid oxidation. The increase in ROS was significantly reduced by vitamin D treatment compared to PA-treated AML-12 cells, and MDA content decreased with vitamin D in a dose-dependent manner. The inflammatory state decreased with vitamin D treatment, as demonstrated by the downregulation of mRNA expression levels of IL-1β, ASC, NLRP3, and TXNIP, which are involved in the activation of pro-inflammatory immune response. Vitamin D also reduced triglyceride overload in hepatocytes and improved lipid metabolism by increasing protein levels of SIRT1, which activated AMPK [67].

### 3.4. Clinical Trials

Starting from these premises regarding the possible role of vitamin D deficiency in NAFLD pathogenesis, one of the first trials on its supplementation in NAFLD patients was published in 2014, when Sharifi et al. showed the effects of vitamin D on serum aminotransferases, IR, oxidative stress, and inflammatory biomarkers. Fifty-three patients with NAFLD were randomly allocated to receive either 50,000 IU vitamin D3 (n = 27) or a placebo (n = 26) every 14 days for 4 months in a double-blind, placebo-controlled study. Various blood chemical parameters of chronic inflammation, oxidative stress, IR, as well as the grade of hepatic steatosis, were evaluated pre- and post-intervention. The conclusions were that an improved vitamin D status in the treated group led to an improvement in patients with NAFLD, but only in serum hs-CRP and MDA [68]. Shortly after, Papapostoli et al. studied 40 patients with vitamin D deficiency and liver steatosis quantified by transient elastography controlled attenuation parameter (CAP) who received 20,000 IU vitamin D/week for six months. They observed that liver steatosis, as assessed by CAP, significantly improved after only 4 weeks of vitamin D correction [69]. 

Another double-blind, single-center trial conducted on patients with NAFLD randomized 201 subjects to receive vitamin D (1000 IU/day) and 110 a matching placebo. Transient elastography (TE, FibroScan), indices of liver steatosis (CAP), and fibrosis (liver stiffness measurement [LSM]), recorded at baseline and after 12 months, showed significant improvements only in those who received the low–medium dose supplementation of vitamin D [70]. On the contrary, a double-blind, placebo-controlled trial performed on a group of T2DM patients with NAFLD randomized to receive either cholecalciferol (26) or placebo (29) for 24 weeks had previously shown no significant improvement of hepatic fat fraction measured by magnetic resonance in the group treated with oral high-dose vitamin D supplementation [71]. A similar conclusion was reached in a study by Amiri et al., which observed no significant effects on anthropometric measures or grade of fatty liver after the administration of 25 µg of calcitriol supplement combined with a weight-loss diet for 12 weeks in 73 NAFLD patients [72]. 

However, a paper on a case study of histology-proven NASH patients showed that receiving 2100 IE vitamin D3 orally over 48 weeks led to a significant improvement of serum ALT levels and a reduction of hepatic steatosis, which was not significant due to the small number of available biopsy specimens, in comparison to the placebo group [73].

Two main studies were also conducted using histological evaluation in a pediatric population. In the first, a combination of docosahexanoic acid (DHA) 500 mg and vitamin D treatment 800 IU was tested in obese children with biopsy-proven NAFLD and vitamin D deficiency in a randomized, double-blind placebo-controlled trial. The authors found an improvement in the NAS in the treatment group, while there was no change in FS, but a reduction of the activation of HSCs and fibrillar collagen content was noted. Moreover, an improvement in transaminases, triglycerides, and homeostasis model assessment of IR (HOMA-IR) was also shown [74]. More recently, El Amrousy et al. conducted a randomized controlled clinical trial on 109 children with biopsy-proven NAFLD, administering either 2000 IU/day vitamin D or placebo for 6 months. What they found was a significant improvement of hepatic steatosis and lobular inflammation, even though there was no significant effect on hepatocyte ballooning or hepatic fibrosis at liver biopsy, so they concluded positively regarding the efficacy of vitamin D supplementation in improving the grades of NAFLD in children [75].

Several other studies have been conducted using indirect markers, such as lipid profiles, liver tests, IR markers, etc., as response indicators. Among them, one reported a significant reduction in fasting plasma glucose, insulin, HOMA-IR, and TG concentrations and an increase in HDL-C in NAFLD patients who received 25 μg calcitriol (n = 37) or 500 mg calcium carbonate + 25 μg calcitriol (n = 37) for 12 weeks, together with following a weight-loss program [76]. Hussain et al. found an increase in vitamin D together with a decrease in HOMA-IR, liver enzymes, and serum CRP in a group of 109 patients with NAFLD who received oral vitamin D3 50,000 IU supplementation weekly for a period of 12 weeks in comparison to a placebo group [77]. A successive study conducted on 100 patients with NAFLD and T2DM demonstrated that in the group receiving conventional therapy plus cholecalciferol for 4 months, there was a significant reduction in serum levels of lipid profile measures, hs-CRP, ALT, STAT-3, NO, hepassocin, and MDA compared to baseline and the placebo group, even though neither group showed significant changes in their glycemic index, TNF-α, AST, or albumin levels [78]. Another study did not find any beneficial effects in terms of serum aminotransferase, alkaline phosphatase, GGT, or lipid profile when comparing vitamin D, calcitriol, and placebo groups at the end of a double-blind, randomized, placebo-controlled trial in patients with NAFLD [79].

Finally, the effects of vitamin D supplementation on NAFLD were also tested in association with other interventions, like fish oil (FO) supplementation (with similar beneficial effects on biomarkers of hepatocellular damage and plasma triacylglycerol levels in FO + vitamin D group, while some suggestive additional benefits compared with the FO only group were detected on insulin levels and inflammation markers) [80], and improvement of lipid metabolism and inflammation markers in NAFLD subjects were found in accordance with elevated serum phosphatidylcholine (16:1/22:6) levels [81] or eccentric exhaustive exercise (EEE) training (vitamin D supplementation reduced liver enzymes and improved lipid profile alterations following EEE in overweight women with NAFLD) [82].

## 4. Silymarin and Its Active Components

### 4.1. Milk Thistle Plant Extracts, a Cornerstone in the Antioxidant Approach to Liver Steatosis

Silymarin, an extract of the milk thistle plant, is composed of six flavonolignans (silybin, isosilybin, silychristin, isosilychristin, silydianin, and silimonin) and other flavonoids, but its principle active component is silybin. 

Silybin, which has been proven to have anti-inflammatory, anti-apoptotic, antioxidant, anti-microbial, and anti-tumor activity, has also been the subject of studies in the field of NAFLD [83,84].

Indeed, an in vivo study of weaned piglets with hepatic oxidative injury induced by Paraquat demonstrated that dietary silybin supplementation increased the mRNA expression levels of superoxide dismutase (SOD), glutathione peroxidase 1 (GPX1), and glutathione peroxidase 4 (GPX4), liver antioxidant enzymes, and decreased malondialdehyde (MDA) and H2O2 levels. Furthermore, silybin counteracted oxidative stress by activating the Nrf2/KEAP1 pathway. The inflammatory response was also influenced by silybin, decreasing the content and mRNA levels of TNF-α and IL-8 in the liver, as well as the level of Nf-κB. Silybin treatment also improved mitochondrial function, restoring the activity of enzymes such as mitochondrial complex I (COXI) and complex V (COX V), modulating redox balance [85].

The antioxidative activity of silybin was shown in in vitro and in vivo models of NAFLD, obtained with palmitate treatment in HepG2 cells and with mice fed a high-fat diet, respectively. After treatment with silybin, ROS and PARP activation levels decreased, though the levels of NAD+ and SIRT1 increased in both models. SIRT1 is an enzyme that activates AMPK, which was increased and consequently inhibited acetyl-CoA carboxylase (ACC) activity. Silybin also inhibited SREBP-1 and one of its targets, fatty acid synthase (FAS), while CPT1a and PPARα were upregulated. This demonstrated the protective effect of silybin treatment regarding hepatic lipids in both in vitro and in vivo models [86]. 

Liu et al. showed the beneficial effects of silybin in a mouse model fed an MCD diet to induce NASH. Silybin (20 mg/kg), administered for 6 weeks, improved oxidative stress; in fact, hepatic MDA was reduced, as were CYP2E1 and CYP4A, two enzymes involved in the production of ROS. Conversely, antioxidant enzymes (CAT, GPx, and HO-1) and the protein expression of Nrf2 were upregulated in silybin-treated mice. Furthermore, silybin treatment improved lipid metabolism, increasing the mRNA levels of genes related to lipid catabolism, such as PPARa or CPT1a, but it did not significantly change the levels of mRNA involved in lipogenesis. Histopathological analysis also showed improvement in ballooning and hepatic steatosis in MCD mice treated with silybin [87].

Another study based on the same in vivo NASH model demonstrated antioxidant and anti-inflammatory effects and improvement in hepatic steatosis by treating mice with silybin (105 mg/kg) for 8 weeks. Silybin decreased CYP2E1 and 4-Hydroxynonenal (4-HNE), increased hepatic SOD and GPx activity, and upregulated antioxidant genes regulated by Nfr2, showing its antioxidant effect. Furthermore, hepatic steatosis and α-SMA levels, and therefore fibrosis, decreased in silybin-treated NASH mice compared to NASH mice. Lipid metabolism improved thanks to the decrease in acetyl-coenzyme A carboxylase alpha (Accα) and the increase in Cpt1a and PPARα caused by silybin. The anti-inflammatory effect was demonstrated by a decrease in the mRNA levels of pro-inflammatory cytokines, such as TNF-α, IL-6, IL-1β, and IL-12β [88].

Interestingly, in mice fed an MCD diet, the administration of silybin (20 mg/kg for 4 weeks) decreased inducible nitric oxide synthase (iNOS) and, consequently, protein nitration, as demonstrated by a low expression of hepatic 3-nitrotyrosine, a biomarker of nitrogen free radical species. Silybin also lowered liver ROS, as well as TBARS levels, reducing oxidative stress. Silybin-treated mice fed an MCD diet showed lower hepatic steatosis and ballooning compared to mice fed an MCD diet and also upregulated mRNA levels of lipid metabolism genes (SCD-1, L-FABP, and AOX) that lower lipotoxicity. The data also showed a decrease in the inflammatory response; in fact, the administration of silybin also reduced NF-kB activity [89].

The combined treatment of silybin and tangeretin (TG) can improve the benefits of silybin by increasing its bioavailability. In a mouse NASH model induced by an MCD diet, experiments showed that silybin treatment, alone or in combination with TG, had a comparable effect in inhibiting the inflammatory response and ameliorating oxidative stress. In fact, silybin administration alone or with TG reduced serum levels of TNF-α and IL-6, increased the levels of oxidative stress markers such as SOD, GSH/GSSG ratio, and decreased MDA. A metabolomic assay was performed and demonstrated that treatment with sibylin and TG promoted the biosynthesis of polyunsaturated fatty acids compared to silybin treatment alone, reducing the expression of genes involved in triglyceride and cholesterol metabolism, lipogenesis, and lipid transport such as peroxisome proliferator-activated receptor gamma (PPARγ), liver X receptor (LXR), fatty acid translocase 36 (CD36). This makes treatment with the two compounds more effective because alterations in the unsaturated fatty acid biosynthesis pathway lead to metabolic disorders, obesity, and other diseases. Despite this, even treatment with sibylin alone had a substantial effect in improving NASH [90].

### 4.2. Clinical Trials

Some preliminary observations were published in 2006 by Federico et al., who consecutively enrolled 59 NAFLD patients and 26 HCV-related chronic hepatitis C patients in combination with NAFLD to be treated with 4 pieces/day of a silybin–vitamin E–phospholipids complex (Realsil (RA); IBI-Lorenzini Pharmaceutical, Italy) for six months, followed by another six months of follow up, in comparison to 32 other patients (20 NAFLD and 12 HCV) who served as a control group (no treatment). After 12 months, US steatosis was significantly improved in the NAFLD-treated group, and liver enzyme levels and hyperinsulinemia showed improvement in treated individuals [91]. The same research group successively performed a randomized controlled trial to assess the efficacy of RA in 179 patients with histologically documented NAFLD, randomizing patients 1:1 to RA or placebo (P) orally twice daily for 12 months; they found that patients receiving RA, but not P, showed significant improvements in liver enzyme plasma levels, HOMA, and liver histology [92]. Similar conclusions regarding the efficacy of silymarin + vitamin E in NAFLD were drawn by Aller et al. in a first study of 36 patients with a diagnosis of NAFLD confirmed by biopsy, who were randomized to be treated with silymarin plus vitamin E and a lifestyle modification for 3 months, or only with a hypocaloric diet. The treatment with silymarin plus vitamin E and a hypocaloric diet ameliorated functioning on hepatic tests and the non-invasive NAFLD index [54]. In a subsequent study, they also analyzed the responses to six months of the same therapy in 54 patients with biopsy-proven NAFLD, characterized according to the PNPLA3 genetic variant status, showing significant statistical changes in the treated patients’ transaminases levels but not in their non-invasive index markers, with a smaller decrease in transaminase levels after treatment with silymarin + vitamin E in G allele carriers than in non-G-allele carriers [93].

Unfortunately, a more recent randomized double-blind placebo-controlled multicenter Phase II trial conducted at five medical centers in the United States on 78 non-cirrhotic patients with NASH, testing a silymarin preparation (Legalon^®^, Rottapharm|Madaus, Mylan) was inconclusive due to the high number of patients who did not meet the entry histological criteria and, most of all, the lack of a statistically significant improvement in the NAS of silymarin-treated patients [94].

However, the final word has yet to be written on the effects of possible antioxidant combination approaches on NAFLD, considering that studies on the joint use of multiple molecules are still being published. One evaluated the effects of the administration of silybin with vitamin D and vitamin E on clinical, metabolic, endothelial dysfunction, oxidative stress parameters, and serological worsening markers in NAFLD patients, finding a statistically significant improvement in all these parameters in the treated group six months after baseline [95] (Figure 2).

## 5. The Role of Lifestyle Modifications

### 5.1. Effects of the Mediterranean Diet on MAFLD: Reducing Liver Damage by Changing Lifestyle Habits

As previously mentioned, to date, there are no pharmacological approaches that have shown a clear benefit in the prevention and/or treatment of MAFLD. Precisely for this reason, both the American [96] and European [97] guidelines recommend lifestyle modifications with the implementation of physical exercise and, above all, changes in eating habits as the truly useful approach for the management of patients with MAFLD. Although various dietary approaches have been evaluated in these patients, the adoption of the Mediterranean Diet (MD) seems to have the biggest impact on patient health. It has been proven that this diet can reduce cardiovascular risk and hepatic fat accumulation, simultaneously improving liver fibrosis [98].

These data are supported by several meta-analyses. Hassani Zadeh S et al. compared three different dietary approaches in ultrasound or biopsy-proven NAFLD patients. Dietary patterns were defined as follows: (1) Western Diet: high intake of processed foods, red meat, refined grains, and high-fat dairy, and low intake of fruits, vegetables, and whole grains; (2) Prudent Diet: high intake of whole grains, fruits, vegetables, low-fat dairy, and poultry, and low intake of fast food, refined grains, and processed foods; (3) MD: according to the classical definition [99]. The authors demonstrated that the Western Diet (n = 8787 subjects) significantly increased the risk of NAFLD development (OR = 1.56, CI = 1.27 to 1.92; *p* ≤ 0.001), whereas both the Prudent Diet (n = 13,023 subjects) (OR = 0.78, CI = 0.71 to 0.85; *p* ≤ 0.001) and the MD (n = 3057 subjects) (OR = 0.77, CI = 0.60 to 0.98; *p* = 0.41) significantly reduced this risk [100].

In another study, Kawaguchi T et al. performed a meta-analysis of six studies investigating the effects of the MD on hepatic steatosis and IR in patients with NAFLD. In their analysis, the authors showed that MD compared to a control diet significantly reduced both FLI (standard mean difference: −1.06; 95% CI: −1.95 to −0.17; *p* = 0.02) and HOMA-IR (standard mean difference: −0.34; 95% CI: −0.65 to −0.03; *p* = 0.03). These results were confirmed by an age-based meta-regression analysis (95% CI: −0.956 to −0.237, *p* = 0.001 and 95% CI: −0.713 to −0.003, *p* = 0.048, respectively, for FLI and HOMA-IR) [101]. Confirming previous results, Haigh L et al. analyzed 26 studies with a total of 3037 subjects who underwent dietary intervention by calorie-restricted diet (n = 9 studies), MD (n = 13 studies), and MD component interventions (n = 4 studies; these studies considered the modification of only one component of MD, i.e., two studies based on EVOO supplementation, one on increased whole grain intake, and one on reduced red meat intake). The authors concluded that both a calorie-restricted diet and the MD had favorable effects on NAFLD patients (reduction of ALT (*p* < 0.001), hepatic steatosis (*p* < 0.001), and liver stiffness (*p* = 0.009) in the calorie-restricted diet, reduction of ALT (*p* = 0.02), FLI (*p* < 0.001), and liver stiffness (*p* = 0.05) in the MD diet), with a strong relationship between the degree of calorie restriction and the degree of positive effects [102]. In this light, another systematic review and meta-analysis compared eight randomized controlled trials (five of which included meta-analyses), proving that MD without energy restriction was able to significantly reduce intrahepatic lipid content (standard mean difference: −0.57; 95% CI: −1.04 to −0.10), but without consistent transaminase modification, compared to a hypocaloric dietary intervention [103].

The beneficial effects of MD on MAFLD appear to be directly related to the ability of this dietary approach to modify many of the metabolic pathways underlying MetS, reducing cardiovascular risk and, overall, both morbidity and mortality of these patients [104,105,106,107,108,109]. Several studies have hypothesized that, although the effectiveness of the MD is nowadays consolidated, the true added value of this dietary approach in the treatment of MetS is connected to the intake of high doses of extra virgin olive oil (EVOO) [110,111].

### 5.2. EVOO and Its Metabolic Effects: A Healthy and Palatable Fat That Modulates Several Metabolic Pathways

EVOO is a food mainly composed of triglycerides (98–99%) with a variable concentration of monounsaturated fatty acids (MUFAs), primarily in the form of omega-9 oleic acid (C18:1), ranging from 55 to 83% of the total fatty acid amount. Other MUFAs are a minority and include omega-7 palmitoleic (C16:1), gadoleic/9-eicosenoic (C20:1, −11), and heptadecenoic acids (C17:1). In addition, EVOO also contains PUFAs such as linoleic acid (C18:2, ω-6) and a-linolenic acid (C18:3, ω-3), with a high ω6/ω3 ratio. EVOO’s lipidic fraction is completed by a low percentage of saturated fatty acids, including myristic acid (C14:0), palmitic acid (C16:0), margaric acid (C17:0), stearic acid (C18:0), arachidic acid (C20:0), behenic acid (C22:0), and lignoceric acid (C24:0). EVOO’s minor compounds, in quantitative terms, are squalene, phytosterols, soluble vitamins (β-carotene and tocopherols), pigments (chlorophyll and carotenes), alcohol triterpene, and polyphenols. The latter includes more than 30 different molecules: secoiridoids (the largest group, including oleacein and oleocanthal, along with aglycone forms of oleuropein and ligstroside), phenolic alcohols (mainly hydroxytyrosol (HT) and tyrosol (Tyr)), phenolic acids, lignans, and flavones. These ‘minor’ compounds are responsible for several of EVOO’s biological properties and sensory attributes (color, odor, flavor, taste, and aftertaste) [110,111].

Several studies have focused on EVOO and its compounds, showing beneficial effects on body weight (quantitative reduction) [112], adipose tissue (quantitative reduction and positive metabolic modulation) [113,114], lipidic ratio (reduction of total cholesterol (TC), TC/high-density lipoproteins (HDL), and low-density lipoprotein (LDL) oxidation; increase of HDL concentrations) [115,116], glycemic metabolism (reduction of T2DM incidence, fasting glucose, and advanced glycosylated end-products, including glycated hemoglobin; activation of the glucagon-like peptide-1 (GLP-1) incretin-pattern) [117,118,119,120], and cardiovascular risk (reduction of LDL oxidation, systolic blood pressure, and cardiovascular events, improvement of nitric oxide (NO)-mediated endothelial function) [121,122,123,124]. To a further extent, but with a lesser degree of evidence (mostly preclinical studies), EVOO intake showed effectiveness in the reduction/modulation of autoimmune [125,126,127,128,129], inflammatory bowel [130,131,132,133], neurodegenerative [134,135,136,137,138], and neoplastic diseases [137,139,140], as well as gut microbiota alterations [141,142,143,144].

Several metabolic pathways have been advocated to explain the health benefits of EVOO. Among these are the upregulation of Nrf2 expression, causing an increase in antioxidant molecule expression [145]; reduction of NF-kB activation, with a consequent downregulation of cytokines, chemokines, adhesion molecules, and inflammatory/oxidative stress enzymes [146,147,148]; activation of phosphoinositide 3-kinase (PI3K/Akt) and mitogen-activated protein kinase (MAPK) (ERK and p38) signaling cascades, thus inducing endothelial NO synthase phosphorylation and modulating NO levels [149]; increase of both MUFA and PUFA absorption, bringing about a reduction of both TC and LDL, decreasing TC/HDL and LDL/HDL ratios [150]; and modification of gut microbiota species and modulation of their metabolism, leading to the modulation of intestinal permeability, antigen exposition, immune cell response, inflammation, and oxidative stress [141,151,152,153].

It is necessary to underline that, despite the large amount of evidence showing an overall beneficial effect of EVOO on human health, there are studies that deny this activity, showing, on the contrary, an absence of effectiveness compared to simple lifestyle modification [154].

### 5.3. EVOO and MAFLD: Fact or Fiction?

In light of EVOO’s positive metabolic effects, which can reduce several MetS components, several research groups have hypothesized that EVOO might reduce liver fat accumulation, oxidation, and inflammation, thus resulting in MAFLD/MASH prevention/regression.

A very recent meta-analysis seems to answer the question of whether or not EVOO provides benefits for MAFLD/MASH. The authors conducted a meta-analysis of seven clinical studies, analyzing multiple biochemical variables and the body mass index (BMI) of a total of 529 subjects, of which 60.9% were men. The intake of olive oil compared to the diets and/or products used as controls did not lead to a significant improvement in any of the biochemical parameters analyzed (including transaminase levels, cholestasis, and lipid metabolism indices) or in the quantity of liver fat, but did result in a significant reduction in BMI (standard mean difference = −0.57 kg/m^2^; 95% CI: −1.08 to −0.06 kg/m^2^, *p* = 0.03). Nonetheless, some points of this study need to be highlighted. The randomized trials analyzed had a high degree of heterogeneity, and the administration of olive oil occurred in different quantities and methods (extracts in capsules, oil to be added to the diet, etc.), as well as associating it with different dietary interventions and comparing it with different substances. Furthermore, the authors generically considered studies in which olive oil and not EVOO was administered, a significant issue if we consider the different composition, both in terms of fats, polyphenols, and other minor compounds, which differentiate an olive oil from an EVOO [155].

The data from the aforementioned meta-analysis led us to hypothesize that only an olive oil with specific chemical features that can identify it as an EVOO [156,157] might have beneficial effects on hepatic metabolism, possibly reducing the degree of steatosis.

However, even when restricting the literature search to the main in vitro animal models and human studies that have specifically analyzed the effects of EVOO, or its components, on NAFLD/MAFLD (Table 1, search string containing the following terms: EVOO and liver steatosis or NAFLD or MAFLD or MASLD), it remains difficult to determine a definitive response regarding EVOO’s effects on liver steatosis [158,159,160,161,162,163,164,165,166,167,168,169,170,171,172,173,174]. In fact, although most of the studies conclude in a substantially positive effect of EVOO on the accumulation of hepatic fat and/or on both systemic and intrahepatic oxidative and inflammatory metabolism, these appear so different from each other in the experimental methodologies used that they can hardly be compared. Noteworthy, almost all studies have been carried out on animal models, and only a limited number of studies have been carried out on humans [172,173,174]. Furthermore, analyzing only the animal models, there were several differences in the animal species used, the methods of inducing liver damage, the quantities of EVOO administered, and the methods of ascertaining liver damage. Finally, in various studies, the exact EVOO composition cannot be deduced, therefore making an overall analysis even more complex. Far from wanting to be an analysis of all the literature present on the effects of EVOO on glycemic, lipidic, cardiovascular, and hepatic metabolism, and assuming that, probably, by widening the search string, various other studies could be found in this regard, it is possible to draw the following conclusions from the previously reported studies:EVOO used in the context of high-fat diets seems to attenuate the harmful effects that these cause on the liver, modulating fat metabolism and reducing its accumulation, preventing (or reducing if already present) hepatic fat content/liver steatosis [172,173,174].EVOO appears to have effects at a systemic level by modulating inflammation and determining a reduction in oxidative damage on tissues [175]; these effects can also be found in the liver, probably reducing inflammatory damage as well as deposition of extracellular matrix components with a consequent reduction in inflammation and liver fibrosis.EVOO appears to be able to act specifically at the level of glucose and lipid metabolism, reducing overall cardiovascular risk and long-term mortality [123,174]. The positive effects on glucose metabolism concern both the production of insulin, with effects at the pancreatic level synergistic with those of GLP-1 [119,120], and the sensitivity of peripheral tissues to the action of insulin [176]. The positive effects on lipid metabolism appear to be linked both to an increase in the quantity of MUFA and PUFA intake, with a consequent reduction in TC, Tg, LDL, and oxidized LDL, and an increase in HDL, and to a quantitative visceral fat reduction, as well as to an optimization of its metabolism [104,116,177].

Considering all this, it might be more correct to state that EVOO appears to have pleiotropic effects on systemic metabolism and that, due to the very close correlation between this and liver fat accumulation, inflammation, and deposition of extracellular tissue, it also exerts hepatoprotective effects. However, many points remain to be clarified. First, whether EVOO is—alone and in the quantities normally consumed in a balanced diet—able to determine the positive effects mentioned above, or whether it represents exclusively one of the components of the MD, and that, outside of this, its nutraceutical value is less [111]. Moreover, we should consider that EVOO is subject, like all foods, to various transformations before its actual use at a systemic level, having to interact with the intestinal microbiota and then undergoing a series of changes during the absorption and metabolism, which could consistently modify its properties tested in vitro and on animal models [178,179,180]. Finally, another point that deserves to be clarified is related to which components of EVOO determine its nutraceutical features (MUFAs vs. PUFAs vs. polyphenols) or whether these are linked to the combination of all its components in the quantities and ratios that are typical of an extra virgin oil. Therefore, it remains absolutely necessary to design randomized and controlled clinical studies which, by specifying a priori the components of EVOO, will be able to defuse all these issues in order to definitively establish whether the putative nutraceutical effects of EVOO are fact or fiction.

## 6. Conclusions

The recent literature is full of studies that focus on non-pharmacological interventions that target oxidative stress in MAFLD treatment. The role of vitamins such as vitamin E and vitamin D, together with silymarin, has been extensively shown to be positive, both in pre-clinical and clinical models. As for the role of EVOO, unfortunately, studies that have specifically analyzed the effects of EVOO or its components on NAFLD/MAFLD have been mostly inconclusive, and thus, it remains difficult to determine a definitive response regarding its effects on liver steatosis. From this point of view, clinical trials designed with rigorous methodology will be necessary to shed more light on this interesting and important topic. 

## Figures and Tables

**Figure 1 antioxidants-13-00731-f001:**
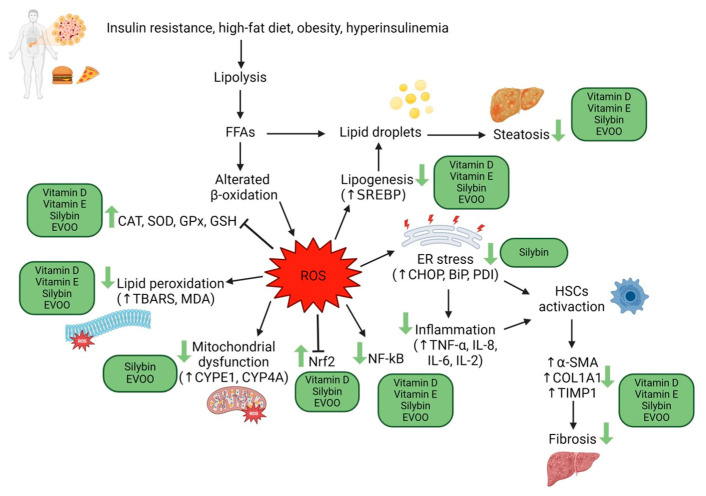
Action mechanisms of oxidative stress in the pathogenesis of liver damage in MAFLD, and possible action sites of vitamins E, D, silymarin, and EVOO. Created with BioRender.com.

**Figure 2 antioxidants-13-00731-f002:**
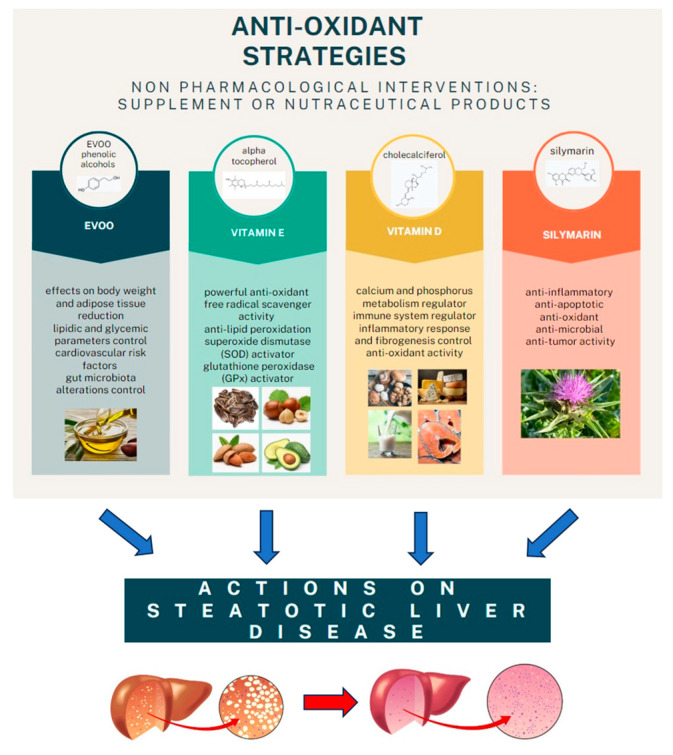
The main anti-oxidant strategies adopted as non-pharmacological interventions in MAFLD.

**Table 1 antioxidants-13-00731-t001:** Main studies analyzing the effects of EVOO on NAFLD/MAFLD/MASLD.

Study Model	Intervention and Used Compounds	Dose	Population	Effect of Treatment	Reference
Cellular model of NASH (palmitic and oleic acid-treated HepG2 cells co-cultured with THP1-derived M1 macrophages and LX2 cells).	Cellular model: co-culture with Tyr at increasing dosages diluted in complete medium.	Cellular model: 0.5, 1, 2, and 5 μM	HepG2 and LX2 cells	Cellular model: reduction of FA accumulation in HepG2 cells and modulation of LX2 activation and macrophage differentiation.	[158]
Mouse model of NASH (high fructose—high-fat diet combined with CCl4 treatment).	Mouse model: Tyr administered daily by oral gavage for 10 weeks.	Mouse model:10 mg/kg	Male C57Bl6 mice	Mouse model: reduction of steatotic and fibrotic areas and hypertrophic hepatocytes without modification of Tg contents. Reduction of proinflammatory cells and reduction of prooxidant enzyme NOX1 and the mRNA expression of TGF-β1 and IL6.	
Iron-induced NAFLD mouse model.	Iron-rich diet (200 mg iron/kg diet) vs. a control diet (50 mg iron/kg diet) with alternate EVOO supplementation for 21 days.	100 mg/day for 21 days	Male Wistar rats	Iron-rich diet-induced liver steatosis, oxidative stress, mitochondrial dysfunction, loss of PUFAs, increasing expression of lipogenic enzymes, and reducing those involved in FA oxidation. EVOO supplementation prevented iron-rich diet effects.	[159]
HFD-induced NAFLD mouse model.	Lard-based HFD vs. EVOO-based HFD vs. phenolic compound-rich EVOO HFD.	EVOO-based HFD: 41% kcals fat from EVOO, 2.92 mg of polyphenols/kg of mouse body weight/day	Female Ldlr−/−. Leiden mice	No differences were proven for liver steatosis. Both EVOO diets improved mice body weight and insulin sensitivity without effects on liver transaminases or increasing LDL liver collagen content. EVOO did not show effectiveness in preventing liver inflammation or fibrosis in gene expression analysis.	[160]
		Phenolic compound-rich EVOO HFD: 41% kcals fat from EVOO, 6.08 mg of polyphenols/kg of mouse body weight/day			
Fish model: spotted seabass juveniles.	Fish model: normal-fat vs. HFD vs. normal fat + HT vs. HFD + HT.	Fish Model: HT 200 mg/kg for 8 weeks	Spotted seabass juveniles	Fish model: HT prevented HFD-increased fat deposition and oxidative stress in the liver.	[161]
Cellular model: zebrafish liver cell line.	Cellular model: Culture with addition of HT, cyclosporin A, and compound C.	HT: 50 µM for 24 h	Zebrafish liver cell line	Cellular model: HT promoted mitochondrial function and activated PINK1-mediated mitophagy. These processes were blocked by both cyclosporin A (mitophagy inhibitors) and compound C (AMPK inhibitor).	
HFD-induced NAFLD mouse model.	Control diet vs. HFD vs. HFD plus n-3 LCPUFA vs. HFD plus EVOO vs. HFD plus n-3 LCPUFA and EVOO.	n-3 LCPUFA 100 mg/kg/die for 12 weeks; EVOO 100 mg/kg/die for 12 weeks	Male C57BL/6J mice	HFD caused liver steatosis (increased total fat, Tg, and free FA contents), glucose and lipid metabolism impairment (glucose, insulin, total cholesterol and Tg serum levels, and HOMA-IR), activation of inflammation (higher TNF-α and IL-6 serum levels), liver and plasma oxidative stress enhancement (decrease of GSH levels), depletion of n-3 LCPUFA hepatic content, increased lipogenic enzyme (ACC and FAS), and reduced lipolytic (CPT-1) enzyme activity. These modifications were either reduced or normalized to control diet values in mice subjected to HFD supplemented with n-3 LCPUFA and EVOO but not in mice subjected to HFD supplemented with n-3 LCPUFA or EVOO alone.	[162]
HFD-induced NAFLD mouse model.	Control diet vs. HFD vs. HFD plus oleacein supplementation.	20 mg/kg for 5 weeks	Male C57BL/6JolaHsd mice	Compared to HFD, mice who received HFD plus oleacein had glucose, cholesterol, and Tg serum levels, as well as liver histology similar to control diet mice. Oleacein positively increased insulin sensibility by modulating protein levels of FAS, SREBP-1, and phospho-ERK in the liver and by reducing body weight.	[163]
Iron=induced NAFLD mouse model.	Iron=rich diet (200 mg iron/kg diet) vs. a control diet (50 mg iron/kg diet) with alternate EVOO supplementation for 21 days.	100 mg/day for 21 days	Male Wistar rats	Compared to control diet, the iron-rich diet increased hepatic total fat, Tg and free FA contents, and serum transaminase levels; in addition, iron-rich diet reduced n-6 and n-3 LCPUFA hepatic and extrahepatic content, increasing n-6/n-3 ratios, and decreasing unsaturation index and Δ5-D and Δ6-D activity. All these changes were prevented by concomitant AR-EVOO supplementation.	[164]
HFD=induced NAFLD mouse model.	Control diet vs. HFD supplemented with DHA or EVOO or DHA + EVOO.	DHA (50mg/kg/day) for 12 weeks; EVOO (50mg/kg/day) for 12 weeks; DHA (50mg/kg/day) + EVOO (50mg/kg/day) for 12 weeks	Male C57BL/6J mice	DHA + EVOO supplementation in HFD mice significantly reduced hepatic steatosis (histologically proven, total fat, Tg, and cholesterol hepatic level reduction), oxidative stress (increase of total serum antioxidant capacity, reduction of liver GSH and 8-isoprostanes levels, and TBARS serum and liver levels), systemic inflammation (TNF-α and IL-6 reduction), and insulin resistance (HOMA-IR reduction) compared to DHA or EVOO supplementation alone. In addition, DHA + EVOO supplementation reduced the activation of lipogenic enzyme (ACC and FAS) and increased it for lipolytic (CPT-1) enzyme.	[165]
HFD=induced NAFLD mouse model.	Control diet vs. HFD supplemented with placebo or water with NO_2_- or EVOO or water with NO_2_- and EVOO	NO_2_- 150 μM; 10% (*w*/*w*) EVOO for 12 weeks; 150 μM NO_2_- and 10% (*w*/*w*) EVOO for 12 weeks	Female C57Bl/6J mice	EVOO consumption reduced body and liver weight; hepatic fat accumulation increased nitro-FA levels in plasma (higher in water with NO_2_- and EVOO group compared with the EVOO alone group) and improved hepatic mitochondrial function (enhancement of both complex II and complex V activity).	[166]
HFD=induced NAFLD mouse model.	Control diet vs. HFD vs. HFD + HOPO vs. HFD + EVOO.	HOPO 10% total fat intake for 12 weeks; EVOO 10% total fat intake for 12 weeks	Male SD rats	Both EVOO and HOPO HFDs reduced body weight gain, HOMA-IR, and liver steatosis (histologically proven, reduction of Tg liver levels) compared to HFD alone. HOPO + HFD reduced total cholesterol, LDL, and Tg serum levels. Both HOPO and EVOO significantly increased microbiota diversity and abundance of Bifidobacterium.	[167]
Osteoarthritis mice model obtained by anterior cruciate ligament transection.	Control diet vs. sicilian EVOO supplemented diet vs. Tunisian EVOO supplemented diet vs. Tunisian EVOO and leaf extract-supplemented diet.	All supplements were provided as 2.5 mL/100 g of chow (i.e., 2.25 g/100 g of chow of EVOO) for 7 days	Male Wistar rats	No differences in liver steatosis were found between groups.	[168]
HFD-induced NAFLD mouse model.	Regular diet vs. regular diet + vitamin D supplementation vs. regular diet + vitamin D restriction vs. HFD-butter + vitamin D supplementation vs. HFD-butter vs. + vitamin D restriction vs. HFD-EVOO + vitamin D supplementation vs. HFD-EVOO vs. + vitamin D restriction	Vitamin D supplementation: 4000 U.I./Kg; Vitamin D restriction: 0 U.I./Kg; HFD-butter and HFD-EVOO 41% energy intake from fats (no more specific data in terms of EVOO supplementation)	Sprague/Dawley male rats	All groups showed a NAFLD activity score between 0 and 2 (not diagnostic of steatohepatitis). Collagen I levels were greater in both HFD-butter + vitamin D supplementation and HFD-butter vs. + vitamin D restriction groups compared to other groups. IL-1 was mostly expressed in all vitamin D-restricted groups. IGF-1 and DKK-1 were reduced in all HFD-butter and HFD-EVOO diets. EVOO supplementation seemed to reduce collagen-1 liver production.	[169]
HFD-induced NAFLD mouse model.	Control diet vs. HFD-lard-based vs. HFD-EVOO-based vs. HFD-based on phenolics-rich EVOO.	Control diet: 13% energy from fat for 24 weeks; HFD-lard-based: 49% energy from fat for 24 weeks; HFD-EVOO-based: 49% energy from fat, 41.7% from EVOO for 24 weeks; HFD-based on phenolics-rich EVOO: 49% energy from fat, 41.7% from EVOO for 24 weeks	C57BL/6J mice	Compared with the HFD-lard-based mice, HFD based on phenolic-rich EVOO reduced total cholesterol and LDL (*p* < 0.001), increasing HDL (*p* < 0.01), whereas EVOO-based HFD reduced Tg (*p* < 0.001). Both EVOO-based diets reduced IFN-γ levels in serum and epididymal adipose tissue, whereas only HFD based on phenolic-rich EVOO reduced IL-6 levels compared to lard-based HFD. Both EVOO-based diets reduced NAFLD activity score, reducing hepatic lipid accumulation (*p* < 0.05) and modulating lipid metabolism (increased PNPLA3 expression (*p* < 0.05)) and composition (increased MUFAs (*p* < 0.05)).	[170]
65HFD-induced NAFLD mouse model.	Control diet vs. control diet + HT vs. HFD vs. HFD + HT	Control diet: 10% fat for 12 weeks; Control diet + HT: 10% fat + 5 mg/kg/day body weight for 12 weeks; HFD: 60% fat for 12 weeks; HFD + HT 60% fat + 5 mg/kg/day body weight for 12 weeks	C57BL/6J mice	HFD determined liver steatosis, inflammation (TNF-α, IL-6, and IL-1β), oxidative stress (GST), depletion of n-3 long-chain PUFAs (26% reduction), downregulation of PPARα and Nrf2, and upregulation of NF-κB. HT supplementation attenuated the metabolic alterations produced by HFD, normalizing the activity of Nrf2, reducing the drop in activity of PPARα, and attenuating the increment in NF-κB activation.	[171]
Prospective randomized human case-control study enrolling patients with high cardiovascular risk but no cardiovascular disease.	MD plus EVOO vs. MD plus mixed nuts vs. low-fat control diet.	EVOO: 1 L/week for 3 years; mixed nuts: 30 g/day (15 g walnuts, 7.5 g hazelnuts, and 7.5 g almonds) for 3 years	100 subjects (63 men aged 55–80 y and 37 women aged 60–80 y); MD enriched with EVOO: 34; MD supplemented with mixed nuts: 36; low-fat control diet: 30	Hepatic steatosis (assessed by magnetic resonance) was present in 8.8%, 33.3%, and 33.3% of the participants in the MD plus EVOO, MD plus nuts, and control diet groups, respectively (*p* = 0.027). Respective mean values of liver fat content were 1.2%, 2.7%, and 4.1% (*p* = 0.07). Median values of urinary 12(S)-hydroxyeicosatetraenoic acid/creatinine concentrations were significantly lower (*p* = 0.001) in the MD plus EVOO group (2.3 ng/mg) than in the MD plus nuts (5.0 ng/mg) and control (3.9 ng/mg) groups. No differences in adiposity or glycemic indexes were proven.	[172]
Prospective observational human study enrolling patients with NAFLD and metabolic syndrome.	MD plus EVOO with high concentration of oleocanthal.	EVOO: 32 g/day for 8 weeks	23 subjects (15 men and 8 women, age: 60 ± 11 years)	Intervention diet, compared to baseline, was associated with a reduction in body weight, waist circumference, BMI, alanine transaminase, and hepatic steatosis (evaluated by fatty liver index), as well as IL-6, IL-17A, TNF-α, and IL-1B, while IL-10 increased. Maximum subcutaneous fat thickness increased, with a concomitant decrease in the ratio of visceral fat layer thickness/subcutaneous fat thickness.	[173]
Prospective cohort human study (1-in-5 random sample study drawn from the electoral list of Castellana Grotte, Italy.	MD plus EVOO at different dosages according to standardized and validated food questionnaire.	EVOO consumption was categorized into four levels: low: <20 g/die; medium–low: 21–30 g/die; medium–high: 31–40 g/die; high: >40 g/die	2754 subjects divided according to EVOO consumption levels: low: 645 (male 340; age 46.38 ± 13.00); medium–low: 635 (male 353, age 52.15 ± 13.99); medium–high: 595 (male 346, age 56.9 ± 14.77); high: 879 (male 522, age 61.77 ± 13.54)	There was a significant negative effect on mortality for the whole sample when EVOO consumption was used, both as a continuous variable and when categorized. The protective effect was stronger in the sub-cohort with NAFLD (778 subjects), especially for the highest levels of EVOO consumption (hazard ratio = 0.58, *p* < 0.05).	[174]

ACC: acetyl-coa carboxylas; AMPK: adenosine monophosphate-activated protein kinase; BMI: body mass index; CPT-1: carnitine palmitoyltransferase 1; DHA: docosahexaenoic acid; DKK: DicKKopf-related protein; ERK: extracellular signal-regulated kinase; EVOO: extra virgin olive oil; FA: fatty acids; FAS: fatty acid synthase; GSH: glutathione; GST: glutathione-S-transferase; HFD: high-fat diet; HOMA-IR: homeostasis model assessment for insulin resistance; HOPO: high oleic acid peanut oil; HT: hydroxytyrosol; IFN: interferon; IGF: insulin-like growth factor; IL: interleukine; LCPUFA: long-chain polyunsaturated fatty acids; LDL: low-density lipoprotein; MAFLD: metabolic dysfunction-associated fatty liver disease; MASLD: metabolic dysfunction-associated steatotic liver disease; MD: Mediterranean diet; MUFAs: monounsaturated fatty acids; NAFLD: non-alcoholic fatty liver disease; Nrf2: nuclear factor erythroid 2–related factor 2; NASH: non-alcoholic steatohepatitis; PNPLA3: patatin-like phospholipase domain-containing protein 3; PPAR: peroxisome proliferator-activated receptor; PUFAs: polyunsaturated fatty acids; SREBP-1: sterol regulatory element-binding protein 1; TBARS: thiobarbituric acid reactive substances; Tg: triglycerides; TNF: tumor necrosis factor; Tyr: tyrosol.

## Data Availability

Not applicable.

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
