# Peer review of "Oxidative Stress as a Target for Non-Pharmacological Intervention in MAFLD: Could There Be a Role for EVOO?"

_antioxidants, 2024, doi:10.3390/antiox13060731_

Round 1

Reviewer 1 Report

The manuscript of Seidita et al. deals with a review article on the therapeutic approaches targeting oxidative stress to treat steatosis, and a number of clinical trials have been conducted to date on subjects with NAFLD using drugs as well as supplements or nutraceutical products, mainly those using Vitamin E, Vitamin D, and Silybin as the most studied substances, as well as non-pharmacological approaches have also been explored, especially lifestyle and diet modifications. Within these, the Mediterranean Diet (MD) seems to be the most reliable for affecting liver steatosis, probably with the added value of the presence of extra virgin olive oil (EVOO). Present data could be useful in the study of gastric cancers, and the manuscript seems to be adequately written. From here, this review could be suitable for publication after addressing some minor concerns:

 1)    Authors say that “At the same time, hyperinsulinemia occurs, increasing lipolysis in peripheral tissues with a consequent increase in free fatty acids” It is not clear for this reviewer how high level of insulin will favor lipolysis instead lipogenesis?

 2)    When listing and discussing those antioxidant enzymes that protect against oxygen toxicity, the role of glutathione reductase should be also commented.

 3)    Authors state that vitamin E is a powerful antioxidant with free radical scavenger activity, avoiding lipid peroxidation, but it also acts by increasing the activity of antioxidant enzymes like SOD and GPx. Since tocopherol is a lipid soluble antioxidant, how can it increase activities of soluble enzymes with antioxidant capabilities?

 4)    The same concerns could be applied to vitamin D, another fat-soluble vitamin.

 5)    Vitamin E could have other properties independent of its antioxidant ability. Could authors discuss what is known in this regard?

The manuscript of Seidita et al. deals with a review article on the therapeutic approaches targeting oxidative stress to treat steatosis, and a number of clinical trials have been conducted to date on subjects with NAFLD using drugs as well as supplements or nutraceutical products, mainly those using Vitamin E, Vitamin D, and Silybin as the most studied substances, as well as non-pharmacological approaches have also been explored, especially lifestyle and diet modifications. Within these, the Mediterranean Diet (MD) seems to be the most reliable for affecting liver steatosis, probably with the added value of the presence of extra virgin olive oil (EVOO). Present data could be useful in the study of gastric cancers, and the manuscript seems to be adequately written. From here, this review could be suitable for publication after addressing some minor concerns: 

1)    Authors say that “At the same time, hyperinsulinemia occurs, increasing lipolysis in peripheral tissues with a consequent increase in free fatty acids” It is not clear for this reviewer how high level of insulin will favor lipolysis instead lipogenesis?

 2)    When listing and discussing those antioxidant enzymes that protect against oxygen toxicity, the role of glutathione reductase should be also commented.

 3)    Authors state that vitamin E is a powerful antioxidant with free radical scavenger activity, avoiding lipid peroxidation, but it also acts by increasing the activity of antioxidant enzymes like SOD and GPx. Since tocopherol is a lipid soluble antioxidant, how can it increase activities of soluble enzymes with antioxidant capabilities?

 4)    The same concerns could be applied to vitamin D, another fat-soluble vitamin.

 5)    Vitamin E could have other properties independent of its antioxidant ability. Could authors discuss what is known in this regard?

Author Response

Comments for Authors

Reviewer #1

1)  Authors say that “At the same time, hyperinsulinemia occurs, increasing lipolysis in peripheral tissues with a consequent increase in free fatty acids” It is not clear for this reviewer how high level of insulin will favor lipolysis instead lipogenesis?

We thank the referee for the correct observation and we hypothesize the following pathophysiological explanation:

“If it is true that insulin increases lipogenesis, being a hormone that stimulates growth, a condition of hyperinsulinemia persistent over time is also linked to insulin resistance, which leads to a decrease in glucose absorption and an increase in lipolysis”, which is corroborated by these studies we added at bibliography:

Choi JH, Gimble JM, Vunjak-Novakovic G, Kaplan DL. Effects of hyperinsulinemia on lipolytic function of three-dimensional adipocyte/endothelial co-cultures. Tissue Eng Part C Methods. 2010 Oct;16(5):1157-65. doi: 10.1089/ten.TEC.2009.0760. PMID: 20144013; PMCID: PMC2943403. 

Fryk E, Olausson J, Mossberg K, Strindberg L, Schmelz M, Brogren H, Gan LM, Piazza S, Provenzani A, Becattini B, Lind L, Solinas G, Jansson PA. Hyperinsulinemia and insulin resistance in the obese may develop as part of a homeostatic response to elevated free fatty acids: A mechanistic case-control and a population-based cohort study. EBioMedicine. 2021 Mar;65:103264. doi: 10.1016/j.ebiom.2021.103264. Epub 2021 Mar 9. PMID: 33712379; PMCID: PMC7992078.

Fernández-Veledo S, Nieto-Vazquez I, de Castro J, Ramos MP, Brüderlein S, Möller P, Lorenzo M. Hyperinsulinemia induces insulin resistance on glucose and lipid metabolism in a human adipocytic cell line: paracrine interaction with myocytes. J Clin Endocrinol Metab. 2008 Jul;93(7):2866-76. doi: 10.1210/jc.2007-2472. Epub 2008 Apr 22. PMID: 18430774.

2)  When listing and discussing those antioxidant enzymes that protect against oxygen toxicity, the role of glutathione reductase should be also commented.

Thanks for the suggestion. A description about glutathione reductase has been added in the text:

“Glutathione reductase also plays a central role in cellular redox homeostasis. Indeed, this enzyme catalyzes the reduction of GSSG to GSH. GSH is one of the most abundant and important antioxidant thiols in humans, modulating ROS and RNS levels.”

Couto N, Wood J, Barber J. The role of glutathione reductase and related enzymes on cellular redox homoeostasis network. Free Radic Biol Med. 2016 Jun;95:27-42. doi: 10.1016/j.freeradbiomed.2016.02.028. Epub 2016 Feb 26. PMID: 26923386.

3)  Authors state that vitamin E is a powerful antioxidant with free radical scavenger activity, avoiding lipid peroxidation, but it also acts by increasing the activity of antioxidant enzymes like SOD and GPx. Since tocopherol is a lipid soluble antioxidant, how can it increase activities of soluble enzymes with antioxidant capabilities?

Fat-soluble vitamins need proteins that solubilize them and allow them to act inside cells. For example, α-tocopherol uses α-tocopherol transfer protein (α-TTP) as a carrier to be transported to liver cells but also to other extrahepatic tissues.

Kono N, Arai H. Intracellular transport of fat-soluble vitamins A and E. Traffic. 2015 Jan;16(1):19-34. doi:10.1111/tra.12231. Epub 2014 Nov 7. PMID: 25262571.

 4)    The same concerns could be applied to vitamin D, another fat-soluble vitamin.

5) Vitamin E could have other properties independent of its antioxidant ability. Could authors discuss what is known in this regard?

As suggested, a description about the other activities of vitamin E have been added in the text:

In addition to antioxidant activity, vitamin E also exerts anti-inflammatory effect by inhibiting COX2 and 5-LOX [Jiang Q. Natural forms of vitamin E: metabolism, antioxidant, and anti-inflammatory activities and their role in disease prevention and therapy. Free Radic Biol Med. 2014 Jul;72:76-90. doi: 10.1016/j.freeradbiomed.2014.03.035. Epub 2014 Apr 3. PMID: 24704972; PMCID: PMC4120831], anti-tumor activity [Abraham A, Kattoor AJ, Saldeen T, Mehta JL. Vitamin E and its anticancer effects. Crit Rev Food Sci Nutr. 2019;59(17):2831-2838. doi: 10.1080/10408398.2018.1474169. Epub 2018 Oct 23. PMID: 29746786], has protective effect against cardiovascular events [Saremi A, Arora R. Vitamin E and cardiovascular disease. Am J Ther. 2010 May-Jun;17(3):e56-65. doi: 10.1097/MJT.0b013e31819cdc9a. PMID: 19451807] and regulates the immune response [Lee GY, Han SN. The Role of Vitamin E in Immunity. Nutrients. 2018 Nov 1;10(11):1614. doi: 10.3390/nu10111614. PMID: 30388871; PMCID: PMC6266234].

Reviewer 2 Report

This narrative review article gives a good overview on non-pharmacological interventions in MASLD treatment, which target oxidative stress such as vitamin E and D, silybin and Mediterranean diet with a focus on the role of EVOO. The article is written in a lively style, which enables a fluent reading despite more of 170 papers, many of them reviews, cited. It gives a concise and precise introduction to the subjects of manner and explains the major open questions. The citations, with one exception (see below), are well and balanced chosen. Also the interpretation of the existing data is balanced, not resulting in an inappropriate jubilation about the positive effect of distinct substances as it may happen in such reviews. In conclusion, the authors stress the necessity of studies with rigorous methodology and sufficient statistical power to investigate the particular clinical effects of the interventions discussed here. I do not see any major issues to be corrected/improved in this article

Minor comments

·         Hepatic stellate cell collagen secretion leading to fibrosis was not originally demonstrated in Ref. 14.

·         Chapter 3 and 4 may benefit from short summaries at their ends.

·         Lines 177-186 are one sentence, hard to read.

·         Line 300. I wouldn´t say the discovery of the nontraditional role of vitamin D happened “recently”.

Author Response

Reviewer #2

Major comments

This narrative review article gives a good overview on non-pharmacological interventions

in MASLD treatment, which target oxidative stress such as vitamin E and D, silybin and

Mediterranean diet with a focus on the role of EVOO. The article is written in a lively style,

which enables a fluent reading despite more of 170 papers, many of them reviews, cited. It

gives a concise and precise introduction to the subjects of manner and explains the major

open questions. The citations, with one exception (see below), are well and balanced

chosen. Also the interpretation of the existing data is balanced, not resulting in an

inappropriate jubilation about the positive effect of distinct substances as it may happen in

such reviews. In conclusion, the authors stress the necessity of studies with rigorous

methodology and sufficient statistical power to investigate the particular clinical effects of

the interventions discussed here. I do not see any major issues to be corrected/improved in this article

Detail comments

As regard as the title, if the Editor agree, it could be changed into: “Oxidative stress as a target for non-pharmacological intervention in MASLD: what's the role for a nutraceutical approach?”

Minor comments

  • Hepatic stellate cell collagen secretion leading to fibrosis was not originally demonstrated in Ref. 14.

As correctly suggested by the referee, another reference explaining the role of HSC in liver figrogenesis was added to the text:

Kisseleva T, Brenner DA. Role of hepatic stellate cells in fibrogenesis and the reversal of fibrosis. J Gastroenterol Hepatol. 2007 Jun;22 Suppl 1:S73-8. doi: 10.1111/j.1440-1746.2006.04658.x. Erratum in: J Gastroenterol Hepatol. 2008 Mar;23(3):501-2. PMID: 17567473.

  • Chapter 3 and 4 may benefit from short summaries at their ends.
  • Lines 177-186 are one sentence, hard to read.

As suggested by the referee the long sentence was split into two shorter (now lines 199-206).

  • Line 300. I wouldn´t say the discovery of the nontraditional role of vitamin D happened “recently”.

As suggested the word “recently” was deleted.